# Toxic Shock Syndrome: A Literature Review

**DOI:** 10.3390/antibiotics13010096

**Published:** 2024-01-18

**Authors:** Enora Atchade, Christian De Tymowski, Nathalie Grall, Sébastien Tanaka, Philippe Montravers

**Affiliations:** 1DMU PARABOL, Bichat-Claude Bernard Hospital, AP-HP, 75018 Paris, France; christian.detymowski@aphp.fr (C.D.T.); sebastien.tanaka@aphp.fr (S.T.); philippe.montravers@aphp.fr (P.M.); 2UFR Diderot, Paris Cité University, 75018 Paris, France; nathalie.grall@aphp.fr; 3INSERM UMR 1149, Immunoreceptor and Renal Immunopathology, Bichat-Claude Bernard Hospital, 75018 Paris, France; 4Bacteriology Department, Bichat Claude Bernard Hospital, AP-HP, Paris Cité University, 75018 Paris, France; 5INSERM UMR 1137 Infection, Antimicrobials, Modelling, Evolution, 75018 Paris, France; 6INSERM, UMR 1188, Diabetes Atherothrombosis Réunion Océan Indien (DéTROI), la Réunion University, 97400 Saint-Denis de la Réunion, France; 7INSERM UMR 1152 ANR 10—LABX-17, Pathophysiology and Epidemiology of Respiratory Diseases, 75018 Paris, France

**Keywords:** exotoxin, *Staphylococcus aureus*, *Streptococcus pyogenes*, toxic shock syndrome toxin 1, staphylococcal enterotoxin, streptococcal pyrogenic exotoxin

## Abstract

Toxic shock syndrome (TSS) is a rare, life-threatening, toxin-mediated infectious process linked, in the vast majority of cases, to toxin-producing strains of *Staphylococcus aureus* or *Streptococcus pyogenes*. The pathophysiology, epidemiology, clinical presentation, microbiological features, management and outcome of TSS are described in this review. Bacterial superantigenic exotoxins induces unconventional polyclonal lymphocyte activation, which leads to rapid shock, multiple organ failure syndrome, and death. The main described superantigenic exotoxins are toxic shock syndrome toxin—1 (TSST-1) and enterotoxins for *Staphylococcus aureus* and *Streptococcal pyrogenic* exotoxins (SpE) A, B, and C and streptococcal superantigen A (SsA) for *Streptococcus pyogenes*. Staphylococcal TSS can be menstrual or nonmenstrual. Streptococcal TSS is linked to a severe group A streptococcal infection and, most frequently, to a necrotizing soft tissue infection. Management of TSS is a medical emergency and relies on early detection, immediate resuscitation, source control and eradication of toxin production, bactericidal antibiotic treatment, and protein synthesis inhibiting antibiotic administration. The interest of polyclonal intravenous immunoglobulin G administration as an adjunctive treatment for TSS requires further evaluation. Scientific literature on TSS mainly consists of observational studies, clinical cases, and in vitro data; although more data on TSS are required, additional studies will be difficult to conduct due to the low incidence of the disease.

## 1. Introduction

Toxic shock syndrome (TSS) is a rare, life-threatening, toxin-mediated infectious process that leads to rapid and severe shock, multiple organ failure syndrome, and death. Its occurrence is linked to the toxin-producing strains of *Staphylococcus aureus* or *Streptococcus pyogenes* (group A streptococcus (GAS)) in a vast majority of cases [1]. However, clinical case reports of TSS involving other bacteria have also been reported [2,3,4,5,6,7,8,9,10].

Scientific literature on TSS mainly consists of observational studies, clinical cases, and in vitro data. The levels of evidence are low, especially when addressing TSS related to pathogens other than *S. aureus* or *S. pyogenes*.

## 2. Methods

This narrative review of the literature was performed using the studies on this topic found in the PubMed database. The keywords used for the manual search were “toxic shock syndrome”, “TSST-1”, “superantigen”, “severe streptococcal infection”, and “necrotizing soft tissue infection”, in the title and in all fields. Original articles, reviews and case reports were all considered. Non-English language articles were excluded.

## 3. Pathophysiology of Toxic Shock Syndrome

The occurrence of TSS is linked to the bacterial secretion of superantigenic exotoxins, which are bacterial virulence factors genetically encoded and secreted. Superantigenic exotoxins are able to induce unconventional activation of T cells by antigen-presenting cells (APCs).

During conventional T-cell activation, the APC absorbs foreign particles, processes protease digestion, and presents them as partially degraded in a specific binding groove in the major histocompatibility complex class II (MHC II), which is expressed on its surface. The Ag-MHC II complex binds to the surface of the T-cell receptor (TCR). This results in monoclonal activation of T cells specific to the antigen (Ag).

In the TSS, the superantigen binds the TCR and MHC II outside the Ag presentation site with high affinity. This results in nonspecific, polyclonal lymphocyte activation of 5 to 30% of the total population of T cells [11,12,13]. This simultaneous polyclonal activation results in a significant activation of NF kappa B, which plays a major role in the generation and expansion of the inflammatory response [1]. This results in a massive release of proinflammatory cytokines, with clinical signs, such as capillary leakage, arterial hypotension, organ failure, and coagulation activation, usually being reported in this setting [1].

Physiopathological specificities of staphylococcal and streptococcal TSS are detailed in the corresponding subparts.

## 4. Staphylococcal Toxic Shock Syndrome

### 4.1. Initial Reports

The first description of this syndrome was published by James Todd and colleagues in *The Lancet* in 1978 [14]. The authors described a pediatric case series of seven children, with clinical presentations including high fever, cephalalgia, confusion, cutaneous rash, conjunctival hyperhemia, and digestive signs. The children progressed to a state of prolonged severe shock associated with renal and hepatic failure and disseminated intravascular coagulation. Exotoxin-producing *S. aureus* was isolated from the foci of infection (empyema and abscess) in two patients and in mucosal swabs (nasopharyngeal, vaginal, and tracheal) in four patients but not from blood, cerebrospinal fluid (CSF), or urine. One patient died, while all the others survived and presented with desquamation of the palm of the hands or sole of the feet during recovery [14]. Staphylococcal TSS in adult patients was then described in the 1980s and predominantly involved menstruating women [15].

### 4.2. Diagnostic Criteria

The diagnostic criteria for staphylococcal TSS were proposed by the Centers for Disease Control and Prevention (CDC) in the 1980s and revised in 2011 [1,16]. These criteria, combining clinical and laboratory aspects, are presented in Table 1.

However, CDC criteria only allow for a retrospective diagnosis, as they include the desquamation of the palms of the hands and soles of the feet, which occurs 8 to 21 days after the onset of the illness [1,12,17]. Moreover, a French multicentric retrospective study describing 102 cases of staphylococcal menstrual TSS (m-TSS) demonstrated that none of them met the CDC criteria for a confirmed TSS, and only half of them met the criteria for a probable TSS [18].

### 4.3. Epidemiology of Staphylococcal Toxic Shock Syndrome

Staphylococcal TSS is rare. According to recent studies, the annual incidence of TSS is estimated to be between 0.03 and 0.07/100,000 population [19,20] and seems to be stable. A peak in incidence (13.7/100,000 persons) was observed in the 1980s in the USA [21], linked to the use of highly absorbent tampons, but its incidence decreased after changes in tampon manufacture [22]. These features explain the differentiation in the literature of staphylococcal TSS between menstrual (m-TSS) and nonmenstrual (nm-TSS) syndromes. While m-TSS cases were largely predominant in the 1980s, compared to nm-TSS cases, the proportion of nm-TSS cases gradually increased over time [19,22]. In the UK, the incidence of m-TSS was estimated to be 0.09/100,000 and that of nm-TSS was estimated to be 0.04/100,000 persons [19]. The highest incidence of m-TSS (1.41/100,000 persons) is observed in women aged 13 to 24 years [23].

### 4.4. Staphylococcal Menstrual TSS

Menstrual toxic shock syndrome (m-TSS), which usually occurs in healthy young menstruating women [23], is linked to vaginal colonization with toxic shock syndrome toxin—1 (TSST-1)-producing *S. aureus* in women without neutralizing antibodies. An American study on a cohort of 262 women showed that between 2003 and 2005, 22.9% of women were vaginally colonized with *S. aureus*, and 4.2% were colonized with TSST-1-producing *S. aureus* during both menstruation and nonmenstruation [24]. The use of tampons creates a physicochemical environment favorable to *S. aureus’* growth and production of TSST-1, in particular by providing oxygen in this anaerobic medium [25]. TSST-1 can bind to vaginal epithelial cells and cross the vaginal mucosa [13,25]. A large majority of m-TSS patients have undetectable levels of protective antibodies at the onset of the illness [26]. In m-TSS, blood culture results are negative in all the publications [16]. As it occurs in the absence of any staphylococcal infection, m-TSS is an exclusively toxin-mediated shock.

In a French multicentric retrospective (2005–2020) study reporting 102 cases of m-TSS, the median age was 18 (16–24) years. No previous comorbid condition was reported in 87% of the cases. Clinical presentation included tachycardia (median heart rate 128 (115–140)/min), high fever (median temperature 39.4 (38.5–40.0) °C), skin rash (87% of the cases), and mucosal involvement (50% of the cases). Digestive signs (abdominal pain, diarrhea, and vomiting) and cephalalgia were very common [18]. Vasopressor support was needed in 84% of the cases, 21% of which needed mechanical ventilation [18]. In this study, all patients were using tampons during their period [18]. However, vaginal cups and intrauterine devices have also been reported in staphylococcal m-TSS [27,28,29,30].

### 4.5. Staphylococcal Nonmenstrual Toxic Shock Syndrome

Nm-TSS can result from any staphylococcal infection with a toxin-producing strain of *S. aureus*. It is most often postoperative, even after relatively simple procedures, but can occur postpartum, after abortion, or because of nonsurgical cutaneous lesions [22]. All types of surgical procedures can precede postoperative TSS, but plastic, orthopedic, and head and neck surgery are most frequently used [12]. Postoperative TSS occurs after a median delay of 4 days after surgery [12]. Blood cultures are positive for *S. aureus* in 50% of cases [31]. As nm-TSS is linked to a staphylococcal infection, it is a mixed septic and toxin-mediated shock.

The clinical presentation of nm-TSS is highly comparable to that of m-TSS, although it occurs in significantly older patients (33 (0–84) years vs. 19 (10–47) years, *p* = 0.008) [31]. A retrospective (2003–2006) multicentric French study compared the clinical presentation of m-TSS (21 cases) and nm-TSS (34 cases). Digestive signs and mucosal involvement were less frequent (74% vs. 100%, *p* = 0.009, and 42% vs. 76%, *p* = 0.024, respectively), but neurological involvement was significantly more frequently observed (61 vs. 29%, *p* = 0.028). No statistically significant difference was observed between m-TSS and nm-TSS in the occurrence of other CDC clinical criteria (fever, rash, desquamation, hypotension, renal, hepatic, and hematologic failure) [31]. Another retrospective (2000–2006) American study compared m-TSS (33 cases) and nm-TSS (28 cases) and reported no difference in clinical presentation [23].

### 4.6. Microbiological Features

The described staphylococcal superantigenic exotoxins include TSST-1 and enterotoxins (of which approximately thirty have been described to date) [32]. TSST-1 is a 194 amino acid protein encoded by the gene *tst* and is responsible for 89 to 95% of m-TSSs and 50% of nm-TSSs, the other half being related to the secretion of staphylococcal enterotoxins A, B, and C [23,25,33,34]. M-TSS are nearly all linked to the USA200 clonal group [25].

*S. aureus* strains are, in a vast majority of cases, methicillin-susceptible strains. In a recent French multicenter retrospective study (2005–2020) describing 102 cases of m-TSS, no case of methicillin-resistant *S. aureus* (MRSA) was described [18]. The UK national surveillance data on 180 TSS (107 nm-TSS) between 2008 and 2012 identified only 7 (3.8%) nm-TSS cases of MRSA isolates [19]. However, 4 cases (7%) of MRSA isolates were identified in an American study (2000–2006) on 61 staphylococcal TSS—2 of them being suggestive of community-associated MRSA and 1 of them being USA400 MRSA [23]. Case reports of nonmenstrual staphylococcal TSS involving MRSA have also been published [35,36,37].

Nasal colonization with TSST-1-producing *S. aureus* could be a risk factor for postoperative TSS. *S. aureus* nasal colonization has been observed in 20 to 80% of the human population [38,39] and identified as a major risk factor for community-acquired and nosocomial infections. A recent prospective multicenter study confirmed that preoperative *S. aureus* carriage in nose, throat or perineum, was associated with both surgical site infections and bloodstream infections [40]. Unfortunately, this study did not provide any information on TSST-1 production by the *S. aureus* isolates. A recent study analyzing nasal colonization of 150 healthy volunteers in Kabul showed that 68.4% of the MRSA isolates were TSST-1 producers. TSST-1 production by MSSA isolates was not reported in this study [41]. To our knowledge, nasal colonization with TSST-1-producing *S. aureus* in patients with staphylococcal TSS have not been specifically assessed. Most of the published studies did not report nasal carriage of *S. aureus*. Celie et al. have reported positive *S. aureus* nasal cultures in some cases of postoperative TSS, but these samples were collected in the operating site in all cases [12]. To our knowledge, the incidence of nasal *S. aureus* colonization in staphylococcal TSS is unknown.

## 5. Streptococcal Toxic Shock Syndrome

### 5.1. Initial Reports

In 1987, Cone et al. described, in the *New England Journal of Medicine*, cases of two patients with severe GAS infection with a clinical presentation similar to staphylococcal TSS. This syndrome was named “streptococcal toxic shock-like syndrome” [42]. Two years later, Stevens et al. reported a case series of 20 patients with severe GAS infection, with clinical presentations including shock, multiorgan system involvement, and rapidly progressive local tissue destruction [43].

### 5.2. Diagnostic Criteria

The diagnostic criteria from the CDC for streptococcal TSS [1,44] are presented in Table 1. They include clinical signs of severity associated with the presence of GAS in a nonsterile site (throat, vagina, and sputum) or a normally sterile site (CSF, blood, peritoneal fluid, and tissue biopsy) [1].

### 5.3. Epidemiology of Streptococcal TSS

Streptococcal TSS is rare. Overall, 8 to 22% of patients with severe *S. pyogenes* infection will develop streptococcal TSS [45,46,47,48]. Approximately 40 to 50% of patients with necrotizing soft tissue infection (NSTI) will develop streptococcal TSS [46,47]. Blood cultures are positive in 60 to 86% of cases [6,49]. Common sources of infection include the vagina, pharyngeal mucosa, skin and soft tissues. Streptococcal TSS can also complicate a minor trauma without skin effraction, pneumonia, intrauterine device, septic arthritis, burn, chickenpox in children, or occur postpartum in young women [46,50]. The source of infection remains unknown in 50% of cases [49].

### 5.4. Clinical Presentation of Streptococcal Toxic Shock Syndrome

Streptococcal TSS mostly occurs in elderly patients between 50 and 69 years of age and in patients with comorbidities (diabetes, malignancy, hepatic disease, chronic renal impairment, and heart disease) [45,48]. Alcoholism and the use of nonsteroidal anti-inflammatory agents (NSAIDs) have also been suspected to be risk factors for streptococcal TSS [46]. A strong association between the use of NSAIDs and occurrence of necrotizing soft tissue infection has been described [51]: there is a 3-fold increased risk for streptococcal TSS [47]. However, the role of NSAIDs in streptococcal TSS remains debated. An experimental study in a murine model showed that NSAIDs’ administration resulted in a 22-fold increase in the number of GAS in an injured muscle [52]. Administration of NSAIDs could also mask the signs of severity of the infection by attenuating inflammatory signs, and delaying the diagnosis with a negative impact of the prognosis.

Clinical presentation of streptococcal TSS was described in a series of 14 cases in North Yorkshire. Hypotension was described in 100% of cases, acute kidney failure in 93%, liver failure in 57%, and disseminated intravascular coagulation in 64% [6]. Multiorgan failure syndrome was reported in 43% of the cases [6]. In this series, streptococcal TSS was associated with a necrotizing infection in 71% of the cases (predominantly NSTI and myonecrosis) [6].

### 5.5. Microbiological Features

The disease occurs after penetration of the exotoxin-producing *S. pyogenes* through a skin or mucous barrier alteration. *S.pyogenes* then spreads to deep tissues. The main superantigenic exotoxins described in *S. pyogenes* are streptococcal pyrogenic exotoxins (SpE) A, B, and C and streptococcal superantigen A (SsA). The majority of streptococcal isolates causing TSS are the *emm1* (41.1% of the cases), *emm3* (8.4% of the cases), *emm28* (8.9% of the cases), and *emm89* (9.8% of the cases) genotypes [45]. Streptococcal TSS occurs more frequently with GAS strains harboring *Spe*A or *Spe*c genes (*p* ≤ 0.001) than those harboring *Ssa genes* [45]. SpeB participates in the rapid dissemination of *S. pyogenes* in the skin and soft tissues in combination with other streptococcal virulence factors, such as soluble M protein, which participate in the local and systemic excessive activation of T lymphocytes, APCs and neutrophils.

## 6. TSS Linked to Other Pathogens

Clinical case reports of TSS involving various bacteria (group B, C and G streptococci, *Yersinia pseudotuberculosis*, *Pseudomonas fluorescens*, *Mycoplasma arthritidis*, *Clostridium*, and coagulase-negative staphylococci (CNS)) have been reported [2,3,4,5,6,7,8,9,10]. The pathophysiology of these probable TSS is not yet well established. Group B and G streptococci produce pyrogenic toxins that are able to induce lethal endotoxin shock in animals [4,5]. Superantigen production has also been documented for *Mycoplasma arthritidis* (*Mycoplasma arthritidis*-derived superantigen) [53] and *Yersinia pseudotuberculosis* [54]. Previous studies have shown contradictory results regarding the ability of the CNS to produce superantigens [55,56,57,58]. However, stimulation of human monocytes by the killed CNS could induce a dose-dependent production of cytokines responsible for the clinical symptoms [59]. To date, only a few cases of these TSS have been reported in the literature, and additional data are therefore needed.

## 7. Management of Toxic Shock Syndrome

### 7.1. Supportive Management

First, it is essential to detect the disease early to start immediate resuscitation [1]. Organ support and symptomatic treatment are similar to any case of septic shock, with hemodynamic optimization (fluid resuscitation and early vasopressor administration), intubation, mechanical ventilation and/or renal replacement therapy being used if needed. There are currently no routine tests to detect the presence of toxins in the blood at the bedside.

### 7.2. Source Control and Eradication of Toxin Production

It is crucial to eradicate the source of toxin production. In the case of m-TSS, the foreign bodies (tampon, intrauterine device, menstrual cup, etc.) must be removed as soon as possible. A vaginal or cervical sample must be collected to detect *S. aureus*. In nm-TSS and streptococcal TSS, a surgical intervention must be urgently performed to explore an operating site, collect bacteriological samples in deep tissues, extensively debride necrotic tissues, drain abscesses, etc.

### 7.3. Bactericidal Antibiotic Treatment

Intravenous bactericidal antibiotics must be administered as soon as possible and within the first hour following suspicion according to the Surviving Sepsis Campaign Guidelines. The steps followed in antimicrobial therapy of TSS are presented in Figure 1.

Empirical antibiotic therapy should target Gram-positive cocci. MRSA should be considered in cases of a high incidence of community-acquired MRSA or evocative patient risk factors, including known carriage, household contamination, previous antibiotic therapy, recent hospital stay, and recent trip in a zone at risk.

In the case of NSTI-associated TSS, infection is frequently polymicrobial [61]. Broad spectrum beta-lactam therapy on Gram-positive cocci, Gram-negative bacilli, and anaerobic strains should be administered immediately without waiting for surgical exploration [60,62]. Multidrug-resistant bacteria must be targeted according to patient risk factors or local ecology [63].

De-escalation of antibiotic therapy is needed after receiving antimicrobial susceptibility testing. The duration of antibiotic treatment has not been evaluated by randomized control studies. Recent data suggest that antibiotic treatment could be safely stopped for 48 to 72 h after the final surgical resection in cases of local and systemic favorable evolution [64,65,66]. The optimal antimicrobial treatment duration in staphylococcal TTS is currently unknown. If new anti-Gram-positive antibiotics (ceftaroline and ceftobiprole) are used, antibacterial with antitoxic activity may be associated.

### 7.4. Adjunctive Therapies

#### 7.4.1. Antitoxic Antibiotics

Until recently, the concept was based on in vitro and experimental data. Protein synthesis-inhibiting antibiotics (clindamycin and linezolid) have in vitro capacities for inhibiting bacterial exotoxin production through the inhibition of the transcription of exotoxin genes.

Clindamycin and linezolid, alone or in combination, have shown a significant inhibitory effect on SpE A production in vitro [67]. Some in vivo observational studies have reported an effect of clindamycin on mortality in severe GAS infections [68,69,70]. Moreover, more than 99% of the *emm1* genotype GAS is susceptible to clindamycin [45]. Adjunction of clindamycin for its inhibitory capacities on protein synthesis to antibacterial treatment is recommended in GAS NSTI [63]. Recently, a retrospective single-center quasi-experimental study did not show any difference in the 30-day mortality rate of 274 NSTI patients receiving linezolid versus clindamycin plus vancomycin in association with the standard Gram-negative and anaerobic antibiotic therapy [71]. All of the studies are observational retrospective studies, and recommendations are based on low levels of evidence.

Staphylococcal sensitivity to clindamycin is variable worldwide, and clindamycin-resistant strains are frequent in some geographical areas [72]. In a case report of staphylococcal TSS, an effect of linezolid administration on TSST-1 production was also mentioned [73].

The optimal duration of antitoxic antibiotics is unknown.

#### 7.4.2. Intravenous Immunoglobulin

In vitro, polyspecific intravenous immunoglobulins G (IVIG) have shown the ability to inhibit the superantigenic activity of streptococcal and staphylococcal exotoxins [74,75]. In vivo, some retrospective observational studies have reported an effect of polyspecific IVIG on mortality during TSS [69]. A European multicentric randomized study investigated the effect of polyspecific IVIG in streptococcal TSS [76]. Unfortunately, this study was stopped prematurely because of insufficient inclusions. A difference in mortality rates was observed (10% in the IVIG group versus 36% in the non-IVIGIV group) but did not reach statistical significance. A significant effect on SOFA score at days 2 and 3 was observed [76]. Another randomized placebo-controlled study evaluated the effect of IVIG in NSTI and did not demonstrate any effect on mortality [77]. More recently, a few meta-analyses were performed, some of them describing an effect of IVIG administration on mortality in streptococcal TSS [78,79], and some of them reporting no effect [80]. In conclusion, the interest in polyclonal IVIG administration as an adjunctive treatment for TSS requires further evaluation. Moreover, possible differences in the efficacy of polyspecific and IgG-based immunoglobulin preparations could be hypothesized but have not been studied so far.

## 8. Outcome of Staphylococcal and Streptococcal TSS

The mortality rate of staphylococcal TSS is estimated to be approximately 5% [19]. Mortality is rare after m-TSS. Contou et al. did not report any deaths in their 102 case series [18]. Other studies reported mortality rates between 0% and 5.7% in m-TSS [19,23]. The mortality rate seems to be significantly higher in nm-TSS, which affects an older and more comorbid population. According to the different studies, it is estimated to be between 4% and 22% [19,23,31]. In a recent review of 96 postoperative TSS (6 streptococcal TSS and the rest staphylococcal TSS), the mortality rate was 9.4%, and 24% of the patients suffered from permanent complications, including additional procedures, amputations, reduced range of motion in a joint, or death [12]. The median duration of hospitalization seems to be significantly higher in nm-TSS (11 (2–50) days versus 5 (2–25) days, *p* < 0.001 in m-TSS) [23].

The mortality rate of streptococcal TSS is high and estimated to be between 14 and 64% in different published series [6,45,47,81]. The lowest mortality rate (<1%) is observed in postpartum streptococcal TSS [45]. An American study describing 3974 cases of invasive GAS infections reported that streptococcal TSS during invasive GAS infection was an independent risk factor for mortality (OR 12.7 (7.5–21.3)) [82]. This observation was confirmed by another European study [47].

## 9. Perspectives on TSS

Two probiotics (*Lactobacillus acidophilus* and *Lacticaseibacillus rhamnosus*) could interfere with *S. aureus* growth and the production of TSST-1 and reduce the incidence of mucosa-associated TSS [83]. In addition, a recombinant TSST-1 variant vaccine demonstrated safety, good tolerance, and immunogenicity in a phase 1 study on 46 healthy adult volunteers [84]. Although interesting, these preliminary data demonstrate the need for extended trials.

## 10. Conclusions

Although TSS is an uncommon pathology, the severity of clinical symptoms, its urgent character and specificities in its management make it essential for clinicians to have good knowledge about it. The pathophysiology, clinical presentation and management of this disease have been widely described. However, most of the data in the scientific literature are provided by retrospective or in vitro studies with a low level of evidence. Further investigations are needed, including evaluation of the role of microbiota and environment in the development of TSS, and therapeutic perspectives assessing the effects of antitoxic antibiotics and their optimal duration, interest of polyclonal IVIG administration, or possible vaccination. However, prospective studies are made difficult by the low incidence of the disease.

## Figures and Tables

**Figure 1 antibiotics-13-00096-f001:**
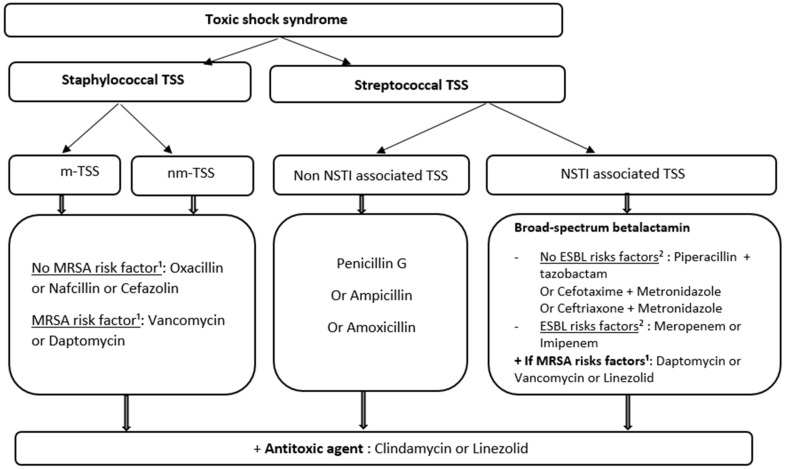
Empiric antimicrobial treatment of TSS [1,60]. TSS: toxic shock syndrome; m-TSS: menstrual TSS; nm-TSS: nonmenstrual TSS; NSTI: necrotizing soft tissue infection; MRSA: methicillin-resistant *Staphylococcus aureus*; ESBL: extended spectrum beta lactamase. ^1^ High incidence of community-acquired MRSA, known carriage, household contamination, previous antibiotic therapy, recent hospital stay, and recent trip in a zone at risk. ^2^ Previous antibiotic treatment in the last 6 months, travel in endemic area in the last 6 months, known carriage of ESBL strain, hospital stay in the last 3 months, and long institution stay.

**Table 1 antibiotics-13-00096-t001:** Diagnostic criteria for staphylococcal and streptococcal TSS according to the CDC recommendations [1,17].

Staphylococcal TSS	Streptococcal TSS
**Clinical criteria**	
Fever ≥ 38.9 °C. Rash—diffuse macular erythroderma. Desquamation—1–2 weeks after onset of the illness, particularly on palms and soles. Hypotension—systolic blood pressure ≤ 90 mm Hg for adults or <5th percentile for children <16 years. Multisystem involvement—at least 3 of the following: a. Gastrointestinal—vomiting or diarrhea; b. Muscular—severe myalgia or elevated creatine phosphokinase twice the upper limit of normal; c. Mucous membranes—hyperhaemia of any mucosal surface; d. Renal—blood urea nitrogen or creatinine twice-upper limit of normal; e. Hepatic—total bilirubin twice-upper limit of normal; f. Hematological—platelets ≤ 100,000/mm^3^; g. Central nervous system-disorientation, combativeness, or alterations in consciousness without focal neurological signs.	Hypotension—systolic blood pressure ≤90 mm Hg in adults or <the fifth percentile by age for children <16 years. Two or more of the following signs: a. Renal impairment: Creatinine greater than or equal to 2 mg/dL (>177 μmol/L) for adults or greater than or equal to twice of the upper limit to normal age. If preexisting renal disease, greater than twofold elevation over the baseline level; b. Coagulopathy—platelets ≤ 100,000/mm^3^ or disseminated intravascular coagulation; c. Hepatic involvement: Alanine aminotransferase, aspartate aminotransferase, or total bilirubin twice the upper limit of normal. If preexisting liver disease, greater than twofold increase over the baseline level; d. ARDS; e. Generalized, erythematous, macular rash that may desquamate; f. Soft-tissue necrosis, including necrotizing fasciitis or myositis, or gangrene.
**Laboratory criteria**	
Negative results on the following tests: Blood, throat or CSF (blood culture may be positive for *S. aureus*); Rise in titer to Rocky Mountain spotted fever, leptospirosis, or measles.	Isolation of group A β-hemolytic streptococci: From a normally sterile site (blood, CSF, joint, pericardial, pleural, peritoneal fluid, tissue biopsy); From a nonsterile site (throat, vagina, sputum).
**Case classification** Probable TSS: a case which meets 4 of the 5 clinical criteria and the laboratory criteria. Confirmed TSS: a case which meets all 5 clinical criteria (including desquamation) and laboratory criteria.	**Case classification** Probable TTS: a case which fulfils clinical case definition and isolation of group A β-hemolytic streptococci from a normally nonsterile site in the absence of other etiology for the illness. Definite TSS: a case which fulfils clinical case definition and isolation of group A β-hemolytic streptococci from a normally sterile site.

TSS: toxic shock syndrome; CSF: cerebrospinal fluid; ARDS: adult respiratory distress syndrome.

## Data Availability

Not applicable.

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
