# Peer review of "Toxic Shock Syndrome: A Literature Review"

_antibiotics, 2024, doi:10.3390/antibiotics13010096_

Round 1

Reviewer 1 Report

Comments and Suggestions for Authors

Well written and very thorough and detailed with up-to-date references

Author Response

Reviewer 1 : Well written and very thorough and detailed with up-to-date references

We would like to thank Reviewer 1 for her/his kind comment.

Reviewer 2 Report

Comments and Suggestions for Authors

The authors have adeptly elucidated the pathogenesis, management, and treatment of Toxic Shock Syndrome (TSS) in this comprehensive review. However, a few minor revisions are warranted to enhance the manuscript:

1. Keyword Section: Eliminate "Toxic shock syndrome" from the keywords, as it is already specified in the title.

2. Italicization of "In vitro": Ensure consistent use of italics for the term "In vitro" throughout the manuscript.

3. Italicization of "and" (Line 24): Remove the italic formatting from the conjunction "and" in line number 24.

4. Methodology Section: Incorporate a dedicated methodology section or paragraph elucidating the database utilized for compiling the literature.

5. Inclusion and Exclusion Criteria: Clearly define the inclusion and exclusion criteria applied in the literature review process.

6. Conclusion Section Enhancement: Strengthen the conclusion by infusing more scientific rigor and inclusivity of future directions. Elaborate on potential areas for further research, advancements, or emerging trends in TSS.

These revisions will refine the manuscript, ensuring a more polished and scientifically robust presentation of the information on Toxic Shock Syndrome.

-------------------

Comments on the Quality of English Language

Minor editing of English language required.

Author Response

Rewiever 2 : The authors have adeptly elucidated the pathogenesis, management, and treatment of Toxic Shock Syndrome (TSS) in this comprehensive review. However, a few minor revisions are warranted to enhance the manuscript:

  1. Keyword Section: Eliminate "Toxic shock syndrome" from the keywords, as it is already specified in the title.

Thank you for your suggestion. We eliminated « Toxic shock syndrome » from the keywords.

  1. Italicization of "In vitro": Ensure consistent use of italics for the term "In vitro" throughout the manuscript.

Thank you for your comment. Italics have been applied to the term « in vitro » as suggested, so as to the term « in vivo ».

  1. Italicization of "and" (Line 24): Remove the italic formatting from the conjunction "and" in line number 24.

We apologize for this mistake. Italic formatting from the conjunction « and » have been removed.

  1. Methodology Section: Incorporate a dedicated methodology section or paragraph elucidating the database utilized for compiling the literature.

Thank you for your comment. This narrative review of the literature was performed using the PubMed databased studies on this topic. Keywords used for the manual search were “toxic shock syndrome”, “TSST-1”, “superantigen”, “severe streptococcal infection”, “necrotizing soft tissue infection”, in title and in all fields. Original articles, reviews, case reports were all considered. According to your suggestion, a method section have been added to the revised manuscript.

  1. Inclusion and Exclusion Criteria: Clearly define the inclusion and exclusion criteria applied in the literature review process.

Thank you for your comment. Original articles, reviews, case reports were all considered. Non-English language articles were excluded. According to your suggestion, this was clarified in the method section of the revised manuscript.

  1. Conclusion Section Enhancement: Strengthen the conclusion by infusing more scientific rigor and inclusivity of future directions. Elaborate on potential areas for further research, advancements, or emerging trends in TSS.

According to your pertinent suggestion, the conclusion has been strengthened, highlighting potential therapeutic perspectives in TSS in the revised version of the manuscript.

These revisions will refine the manuscript, ensuring a more polished and scientifically robust presentation of the information on Toxic Shock Syndrome.

Reviewer 3 Report

Comments and Suggestions for Authors

This is an excellent narrative review of toxic shock syndrome, a rare and life-threatening infectious syndrome due to staphylococcal  and streptococcal exotoxins. It is well-written, while it covers all the important aspects of pathogenesis, diagnosis, and treatment. However, there are some minor issues that could be addressed prior to consideration for publication.

1) The literature search is not described. Please provide a list of terms used for the literature search in the relevant medical databases.

2) Please discuss the influence of nasal colonization with a TSST-1-producing Staphylococcus aureus on the syndrome risk.

3) Please discuss what is the role of NSAIDs in increasing the risk of the syndrome.

4) Regarding antitoxin treatment, linezolid or clindimycin can be equally effective. However, clinical efficacy data is not given. The recent Dorazio quasi-experimental study (Open Forum Infectious Diseases 2023) provides such data concerning the closely related necrotizing soft tissue infections.

5) The immunoglobulin issue seems unresolved. Please discuss the possible differences in efficacy between the polyspecific and the IgG-based immunoglobulin preparations.

Author Response

Rewiever 3 : This is an excellent narrative review of toxic shock syndrome, a rare and life-threatening infectious syndrome due to staphylococcal  and streptococcal exotoxins. It is well-written, while it covers all the important aspects of pathogenesis, diagnosis, and treatment. However, there are some minor issues that could be addressed prior to consideration for publication.

1) The literature search is not described. Please provide a list of terms used for the literature search in the relevant medical databases.

Thank you for your comment. This narrative review of the literature was performed using the PubMed databased studies on this topic. Keywords used for the manual search were “toxic shock syndrome”, “TSST-1”, “superantigen”, “severe streptococcal infection”, “necrotizing soft tissue infection”, in title and in all fields. Original articles, reviews, case reports were all considered. Non-English language articles were excluded. According to your suggestion, a method section have been added to the revised manuscript.

2) Please discuss the influence of nasal colonization with a TSST-1-producing Staphylococcus aureus on the syndrome risk.

Thank you for your suggestion.

Indeed, nasal colonization with TSST-1 producing S. aureus could be a risk factor for postoperative TSS. S. aureus nasal colonization is observed in 20 to 80% of the human population (Brown, Front immunol, Sakr, Front immunol 2018), and has been identified as a major risk factor for community-acquired and nosocomial infections. A recent prospective multicenter study confirmed that preoperative S.aureus carriage in nose, throat or perineum, was associated with both surgical site infections and bloodstream infections (Troeman, JAMA, october 2023). Unfortunately, this study did not bring any information on TSST-1 production by the S. aureus isolates.  Little is known about TSST-1 production of S.aureus in nasal swabs. A recent study (Naimi et al., Journal of global antimicrobial resistance, 2023) analyzing nasal colonization of 150 healthy volunteers in Kabul showed that 68.4% of the MRSA isolates were TSST-1 producers.  TSST-1 production of MSSA isolates was not reported in this study.

To our knowledge, nasal colonization with TSST-1 producing S. aureus in patients with staphylococcal TSS has not been specifically assessed. Nasal carriage of S. aureus was not mentioned in most of the studies. Celie et al. have reported positive S. aureus nasal cultures in some cases of post-operative TSS, but these samples were collected in the operating site in all the cases (Celie, Plast Reconstr Surg Open, 2020). To our knowledge, the incidence of nasal S. aureus colonization in staphylococcal TSS is unknown.

These precisions have been added in the revised version of the manuscript.

3) Please discuss what is the role of NSAIDs in increasing the risk of the syndrome.

Thank you for your suggestion. A strong association between the use of NSAIDs and occurrence of necrotizing soft tissue infection has been described (Souyri, clinical and experimental dermatology, 2008). In the Lamagni et al. study, the authors have reported a 3-fold increased risk for streptococcal TSS in patient who used NSAIDs (Lamagni et al., emerging infectious disease, 2008). However, their role remains debated. An experimental study in a murine model showed that the administration of NSAIDs would result in a 22-fold increase in the number of GAS in an injured muscle (Hamilton et al., the journal of infectious disease, 2008). The administration of NSAIDs could also mask the severity signs of infection by attenuating inflammatory signs and delaying the diagnosis with a negative impact of the prognosis.

These hypothesis about the role of NSAIDs in streptococcal TSS have been clarified in the revised version of the manuscript.

4) Regarding antitoxin treatment, linezolid or clindimycin can be equally effective. However, clinical efficacy data is not given. The recent Dorazio quasi-experimental study (Open Forum Infectious Diseases 2023) provides such data concerning the closely related necrotizing soft tissue infections.

Thank you for your comment. Indeed, this recent retrospective single-center quasi-experimental study did not show any difference in 30-day mortality rate among 274 NSTI patients receiving linezolid versus clindamycin plus vancomycin in association with standard Gram-negative and anaerobic antibiotic therapy. These results have been added to the revised version of the manuscript.

5) The immunoglobulin issue seems unresolved. Please discuss the possible differences in efficacy between the polyspecific and the IgG-based immunoglobulin preparations.

Thank you for your pertinent question. Prospective randomized placebo-controlled studies on this topic (Madsen et al., Darenberg et al.) have assessed the effect of polyspecific intravenous immunoglobulin administration on mortality. Some retrospectives studies did not specify which type of immunoglobulin preparation was administered. In the absence of studies on IgG-based immunoglobulin preparations, this question cannot be answered. This point was mentioned in the revised version of the manuscript.